# A Proximity Complementation Assay to Identify Small Molecules That Enhance the Traffic of ABCA4 Misfolding Variants

**DOI:** 10.3390/ijms25084521

**Published:** 2024-04-20

**Authors:** Davide Piccolo, Christina Zarouchlioti, James Bellingham, Rosellina Guarascio, Kalliopi Ziaka, Robert S. Molday, Michael E. Cheetham

**Affiliations:** 1UCL Institute of Ophthalmology, 11-43 Bath Street, London EC1V 9EL, UK; d.piccolo@ucl.ac.uk (D.P.); christina.zarouchlioti@ucl.ac.uk (C.Z.); rosellina.guarascio.13@ucl.ac.uk (R.G.); kalliopi.ziaka.16@ucl.ac.uk (K.Z.); 2Department of Biochemistry and Molecular Biology, University of British Columbia, Vancouver, BC V6T 1Z3, Canada; rober.molday@ubc.ca

**Keywords:** ABCA4-related retinopathy, Stargardt disease, missense variants, protein misfolding, folding correctors, protein mistrafficking, plasma membrane protein, ABCA4

## Abstract

ABCA4-related retinopathy is the most common inherited Mendelian eye disorder worldwide, caused by biallelic variants in the ATP-binding cassette transporter ABCA4. To date, over 2200 ABCA4 variants have been identified, including missense, nonsense, indels, splice site and deep intronic defects. Notably, more than 60% are missense variants that can lead to protein misfolding, mistrafficking and degradation. Currently no approved therapies target ABCA4. In this study, we demonstrate that ABCA4 misfolding variants are temperature-sensitive and reduced temperature growth (30 °C) improves their traffic to the plasma membrane, suggesting the folding of these variants could be rescuable. Consequently, an in vitro platform was developed for the rapid and robust detection of ABCA4 traffic to the plasma membrane in transiently transfected cells. The system was used to assess selected candidate small molecules that were reported to improve the folding or traffic of other ABC transporters. Two candidates, 4-PBA and AICAR, were identified and validated for their ability to enhance both wild-type ABCA4 and variant trafficking to the cell surface in cell culture. We envision that this platform could serve as a primary screen for more sophisticated in vitro testing, enabling the discovery of breakthrough agents to rescue ABCA4 protein defects and mitigate ABCA4-related retinopathy.

## 1. Introduction

ABCA4-related retinopathy is the most common inherited Mendelian eye disease in the world, with an estimated prevalence of 1:6578 [1]. In the ‘classic’ form, known as Stargardt disease (STGD1, OMIM 248200), central vision is usually lost between adolescence and young adulthood. However, the disease can manifest in a wide variety of clinical forms, from fast and aggressive childhood-onset cone–rod dystrophy and retinitis pigmentosa-like phenotypes, to a mild, late-onset macular pattern sometimes confused with age-related macular degeneration [2]. Despite being such a prominent part of the inherited retinal disease landscape, there are no current approved treatments. The ABCA4 gene encodes for the ATP-binding cassette sub-family A member 4 (ABCA4) protein, which mainly localises in the outer segment (OS) disc rims of photoreceptor (PR) cells [3,4,5,6]. ABCA4 is an important component in the visual cycle, facilitating the clearance of toxic retinoid metabolites. Loss of ABCA4 function will lead to the accumulation in the retinal pigment epithelium (RPE) of lipofuscin fluorophore A2E, a byproduct of the visual cycle, causing the death of RPE cells and, subsequently, the death of PRs [7,8,9,10,11]. 

Over 2200 disease-causing variants have been identified in ABCA4, including missense, nonsense, indels, splicing defects, deep-intronic and structural variants [12], all leading to the loss, or reduced, function of ABCA4 protein. More than 60% are missense variants and can be found throughout the amino acid sequence. These are thought to affect the correct protein folding, traffic, and/or function of the protein [2]. 

ABCA4 missense variants have been mainly studied in heterologous expression systems to investigate the localisation and function of purified proteins. Wild-type (WT)-ABCA4 predominantly localises to vesicular structures, whereas many missense misfolded variants do not traffic to these structures and are retained in the endoplasmic reticulum (ER) [13,14,15,16,17,18]. Missense variants in the cytoplasmic and extracellular domains, as well as in transmembrane domains, have been characterised in terms of localisation and functional activity [14]. 

For missense variants that misfold, restoring ABCA4 protein function could be achieved through a therapeutic approach based on folding correction using chemical chaperones, pharmacological chaperones, or altering molecular chaperone activity. These approaches can rescue defects in the folding of mutant proteins by providing stabilisation during the folding of the protein, thereby enabling the correct conformational maturation. Chemical chaperones, or kosmotropes, display a non-specific mode of action, stabilising protein structure and assisting folding, or enabling the mutant proteins to escape the cell quality control systems [19,20,21]. Pharmacological chaperones are low molecular weight compounds that directly bind to the target protein, promoting thermodynamic stabilisation and template-based induction of proper folding, correcting aberrant folding and thereby enhancing protein function [22,23]. On the other hand, molecular chaperones in the cell assist the folding process through reducing unwanted interactions and stabilise folding intermediates, without being a part of the final structure [24,25,26].

A major advance in the field of small molecule and protein repair has been achieved for transmembrane conductance regulator (CFTR), also known as ABCC7 [27]. Mutations in CFTR cause cystic fibrosis, the most common lethal autosomal recessive disorder in the Caucasian population [28]. Using a high-throughput screening (HTS) cell-based fluorescence membrane potential assay [29], various modulators have been identified and subsequentially characterised, like the correctors VX-809 [30], VX-661 and VX-445 [31]. The combination of these compounds with the potentiator VX-771 [32] has culminated in the approval of four different drug cocktails to treat nearly 90% of patients affected by cystic fibrosis [33] (https://www.cff.org/, accessed on 17 April 2024).

Plasma membrane proteins are the most fruitful HTS targets, with 60% of the approved drugs targeting these biomolecules [34]. Besides CFTR, examples can be found in many other proteins, like Rhodopsin (RHO) [35], ABCB1 [36], ABCC1 and ABCG2 [37]. Moreover, missense variants in different ABC transporters have been investigated for their potential to be rescued by small molecules in terms of folding, traffic and function, like ABCA1 [38], ABCA3 [39] and ABCA4 itself [40,41]. 

Here we describe a novel in vitro platform designed to rapidly and reliably screen ABCA4 traffic to the plasma membrane, serving as a surrogate marker for rescuing the misfolding of missense variants. This platform could be adapted for high-throughput screening. Through the utilisation of this innovative cell-based assay, we identified two potential ‘hit’ compounds, 4-Phenylbutyric acid (4-PBA) and 5-Aminoimidazole-4-carboxamide ribonucleotide (AICAR), from a candidate molecules screen. These compounds demonstrated the capability to enhance ABCA4 traffic to the cell surface, and their effects were validated through complementary approaches.

## 2. Results

### 2.1. ABCA4 Missense Variants Are Retained in the ER

We selected a group of missense variants from different domains of the ABCA4 protein: T983A, A1038V and R1108C in the nucleotide binding domain (NBD) 1 and R2077W in NBD2 (Figure 1). 

WT-ABCA4 and the missense variants T983A, A1038V, R1108C and R2077W were expressed in CHO cells following transient transfection. Between 20–40% of the cells were positive for ABCA4 and no major differences were observed in transfection efficiency between the WT and missense variants. All the ABCA4 variants were tagged with the 1D4 epitope derived from the C-terminus of bovine RHO [43], which has been reported not to alter the protein function and localisation [16]. These misfolding variants tend to localise in a reticular pattern that is characteristic of ER labelling [18,44]. In accordance with the published data, WT-ABCA4 predominantly localised in large intracellular vesicles, which were also positive for the ER marker calnexin (CNX); however, a fraction was also distributed in a reticular pattern characteristic of the ER [16] (Figure 2). In contrast, almost no vesicular-like structures were observed for T983A A1038V, R1108C and R2077W variants, all co-localising with CNX in an ER-like pattern (Figure 2). Therefore, the immunofluorescence data suggest that these variants are retained in the ER by the quality control system of the cell as misfolded protein.

### 2.2. ABCA4 Missense Variants Do Not Traffic to the Plasma Membrane in Cultured Cells

ABCA4 is a membrane protein of the OS of PRs; however, cultured cells do not have an OS and some OS membrane proteins, like RHO [45], are localised predominantly on the plasma membrane when heterologously expressed. ABCA4 has been reported to localise to the cell surface in HEK293T cells [40,46], and in RPE cells derived from iPSCs [47,48]. Therefore, the plasma membrane localisation of WT-ABCA4 and missense variants was investigated. A protocol was developed to discriminate between ABCA4 present on the cell surface and the intracellular protein fraction. At 48 h post transfection, live cells were incubated with an antibody recognising an extracellular epitope (Abbexa, Gly1398-Asn1727) without permeabilization, thereby staining only ABCA4 with the epitope exposed to the extracellular fluid at the plasma membrane. Following this step, cells were fixed, permeabilised and incubated with the 1D4 antibody to stain the total intracellular protein fraction. Whilst there was variability between cells expressing WT-ABCA4 protein, ABCA4 was detectable on the plasma membrane in some cells. In contrast, missense ABCA4 variants showed a much lower level of cells positive for membrane staining (Figure 3A). These data suggest that a fraction of WT-ABCA4 can traffic to the plasma membrane in a heterologous expression system and that the missense variants are less likely to be trafficked to this cellular compartment. 

The variants T983A and R2077W were chosen for further experiments as representative variants in the nucleotide binding domain 1 (NBD1) and nucleotide binding domain 2 (NBD2), respectively (Figure 1). Biochemical investigation of ABCA4 traffic was performed using N-glycosidase F (PNGase F) and Endoglycosidase H (Endo H) to assess glycosylation status. Once in the Golgi, several modifications occur to the glycans of glycosylated proteins and they become resistant to Endo H, but remain sensitive to PNGase F [49]. ABCA4 from mouse retinas showed only a slight increase in mobility after glycosidase treatment (Appendix A), in accordance with previously published data using the bovine protein [3] and the mouse protein [50]. Heterologous ABCA4 expressed in HEK293T cells migrated as two immunoreactive bands (referred to from now on as band A and band B), suggesting two different protein species. The treatment with the glycosidases changed the mobility of band B, indicating that this protein form was glycosylated, whereas band A was resistant to the glycosidases, suggesting that it is the immature, non-glycosylated protein (Figure 3B). Moreover, the T983A and R2077W missense variant B/A band ratios were ≥50% reduced compared to WT-ABCA4 (Figure 3C). These data confirm that ABCA4 missense variants have altered glycosylation status. We further investigated WT-ABCA4, T983A and R2077W traffic by co-staining with the Golgi marker GM130 and no co-localisation was detected (Appendix A).

### 2.3. Reduced Temperature Increases ABCA4 Protein Level and Promotes Traffic to the Cell Surface

Lowering the growth temperature can positively impact the folding of misfolded proteins. For example, temperature-sensitive ΔF508-CFTR can exit the ER, acquire Golgi-specific glycosylation features and function as a channel on the cell surface when the cells are grown at reduced temperatures [51,52,53]. We investigated whether WT-ABCA4 and missense variant traffic and folding were also temperature-sensitive and thereby potentially rescuable pharmacologically. The steady-state protein levels of WT-ABCA4 and T983A, A1038V, R1108C and R2077W variants were increased when transfected HEK293T cells were grown for 48 h at a reduced temperature of 30 °C instead of 37 °C (Figure 4A,B), indicating a potential thermodynamic stabilising effect on ABCA4 produced by the lowering of the temperature and/or reduced degradation. 

To test whether the mutants were rapidly degraded, cells were treated with cycloheximide, an inhibitor of protein translation. WT-ABCA4 and R2077W protein levels showed a slow rate of degradation, whereas T983A, A1038V and R1108C variants expression levels were diminished by ≥50% over a 4 h period, indicating that those mutants were degraded faster than the WT or R2077W. Lowering the temperature to 30 °C improved the stability over time of all the mutants, except for R1108C (Appendix A). 

The traffic to the cell surface of ABCA4 was also investigated using confocal microscopy after growth at the lower temperature. The number of cells with WT-ABCA4 detectable at the plasma membrane increased at 30 °C compared to 37 °C, and a similar effect was also observed for the T983A, A1038V, R1108C and R2077W variants (Figure 4C,D). An increase in the intracellular staining and the vesicular structures of ABCA4 variants was also observed at 30 °C compared to 37 °C (Figure 4C,D), suggesting that lower growth temperature can assist ABCA4 folding and can help the misfolded variants escape the ER and traffic either to the plasma membrane or to vesicular-like structures. In summary, these data show that lowering the temperature can have a stabilising effect on ABCA4 protein level, stability and localisation. Furthermore, they suggest that the folding of these missense variants is rescuable. 

### 2.4. Split NanoBit Proximity Complementation Assay: A New HTS-Friendly Platform to Study Protein Localisation to the Plasma Membrane

We designed a new luciferase assay able to quickly and quantitively monitor ABCA4 plasma membrane localisation, and RHO was used as a membrane protein control. The technology is based on a protein complementation assay system, the NanoLuc Binary Technology system, a two-luciferase-subunit system used to detect protein–protein interactions [54]. In contrast to the use of this system for protein–protein interactions, here the system is adapted for proximity, driving the complementation of the subunits. The Large Bit (LgBiT) subunit was fused to the C-terminus of the first 15 amino acids of the RP2 protein. RP2 is a GTPase-activating protein for ARL3, which was previously shown to be dual acylated at the N-terminus, leading this protein to localise to the plasma membrane when expressed in cells [55]. When the RP2 N-terminal motif is fused to a protein of interest, it causes the fusion protein to localise on the cytoplasmic face of the plasma membrane. The Small Bit (SmBiT) subunit was fused to the C-terminal cytoplasmic domains of ABCA4 and RHO. If the two complexes (RP2-LgBiT and ABCA4-SmBiT or RHO-SmBiT) are both at the cell surface, then the fragments combine and functional luciferase is generated. Traffic to the cell surface can then be followed in real time using luminescence of live cells (Figure 5A). 

Plasmids encoding for WT-ABCA4-SmBiT, T983A-SmBiT, A1038V-SmBiT, R1108C-SmBiT, R2077W-SmBiT, WT-RHO-SmBiT, P23H-RHO-SmBiT and RP2-LgBiT (Appendix A) were produced and their localisation was assessed by immunofluorescence in HEK293T cells (Figure 5B–F). WT-RHO-SmBiT staining delineated the cell borders indicating localisation at the plasma membrane, whereas P23H-SmBiT was resident in the ER, as previously reported [56]. RP2-LgBiT was also confirmed to have a staining pattern corresponding to the cytoplasmic face of the plasma membrane. WT-ABCA4-SmBiT was found to localise both in intracellular vesicles and at the plasma membrane. On the other hand, ABCA4-SmBiT missense variants showed an ER-like pattern. 

The luciferase system was tested using RP2-LgBiT and WT-RHO-SmBiT or P23H-RHO-SmBiT, as positive and negative controls for plasma membrane traffic, respectively. There was a strong luminescence signal produced by the WT-RHO-SmBiT + RP2-LgBiT compared to the P23H-RHO-SmBiT + RP2-LgBiT combination (Figure 5G), establishing that the assay can be used to discriminate between variants trapped in the ER from correctly folded protein reaching the plasma membrane. The WT-ABCA4-SmBiT + RP2-LgBiT plasmids also produced a strong luminescence signal when compared to non-transfected cells (Appendix A). 

These data suggest that the fusion proteins of interest are trafficked as expected and that their proximity is sufficient to enable NanoLuc complementation and produce luminescence. The traffic of the T983A, A1038V, R1108C and R2077W ABCA4 missense variants was compared to the WT-ABCA4 protein using the NanoBit platform. There was no substantial decrease in protein levels among the different variants, such that any difference in luminescence would be related to protein localisation and not protein levels (Appendix A). A significantly lower luminescence level was observed for the T983A-SmBiT + RP2-LgBiT, R1108C-SmBiT + RP2-LgBiT, R2077W-SmBiT + RP2-LgBiT compared to WT-ABCA4-SmBiT + RP2-LgBiT, indicating impaired traffic to the cell surface (Figure 5H). 

The A1038V-SmBiT + RP2-LgBiT combination produced luminescence levels similar to the WT-ABCA4-SmBiT+ RP2-LgBiT combination. Interestingly, the A1038V variant in isolation has been shown to traffic to the OS of PRs in *Xenopus laevis* [57], suggesting that this assay more accurately predicts traffic in PRs than the other cell-based assays we used. These data confirm that some ABCA4 missense variants do not traffic effectively to the plasma membrane. 

Given the effect of reduced growth temperature on ABCA4 protein traffic, traffic was investigated after growth at 30 °C. Surprisingly, the luminescence signal of WT-RHO and WT-ABCA4 + RP2-LgBiT was greatly reduced at 30 °C compared to 37 °C (Appendix A). We therefore tested the control interacting proteins in the unmodified NanoLuc Binary Technology system used at 30 °C, which also showed a sharp decrease in luminescence signal (Appendix A), suggesting the luciferase complementation fails at reduced temperatures.

### 2.5. The Proximity Complementation System Can Detect Pharmacological Rescue

To obtain proof of concept that the system could detect pharmacological rescue of cell surface traffic, we used the P23H-RHO + RP2-LgBiT. As mentioned before, P23H-RHO is retained by the quality control machinery in the ER and is not trafficked to the plasma membrane. The traffic of P23H-RHO to the cell surface can be restored after treatment with 11-cis or 9-cis-retinal, inverse agonists that stabilise RHO structure and improve P23H-RHO traffic [56,58,59]. A dose-dependent increase in luminescence signal was observed following treatment with the 9-cis-retinal in cells transfected with the P23H-RHO + RP2-LgBiT (Figure 6A), confirming that the Split NanoBit system can identify increased plasma membrane luminescence as a consequence of pharmacological rescue of misfolding variants. 

### 2.6. 4-PBA and AICAR Can Promote ABCA4 Traffic to the Plasma Membrane

ABCA4 is a member of the family of ATP-binding cassette transporters (ABC transporters). Small molecules have been reported to rescue the traffic of several ABC proteins, like ABCA1 [38], ABCA3 [39] and CFTR [60]. A candidate small molecule screen on ABCA4 traffic to the plasma membrane was performed based on previously published correctors of ABC transporters. This included testing for the CFTR correctors VX-809 and VX-661, the kosmotropes 4-PBA and TMAO and the AMPK activators Metformin and AICAR. Of these compounds tested, 4-PBA and AICAR showed a positive dose response effect of increased luminescence signal for WT-ABCA4 and T983A, A1038V, R1108C and R2077W variants (Figure 6B,C). By contrast, no statistically significant improvements were obtained following VX-809, VX-661, TMAO or Metformin treatment on the WT-ABCA4 or the variants (Appendix A). 

The effect of these compounds on ABCA4 protein was also confirmed by western blotting, with an increase in the steady-state levels of proteins observed, similar to thermodynamic rescue observed at 30 °C, with both 4-PBA (Figure 7A,B) and AICAR (Figure 8A,B). Moreover, immunofluorescence using the sequential antibody staining described above confirmed the positive effect of 4-PBA (Figure 7C) and AICAR (Figure 8C) on ABCA4 trafficking to the cell surface. Whilst we did not observe the T983A variant on the cell surface following AICAR treatment, an increase in the vesicular ABCA4 protein staining for WT and variants was observed following treatment with both drugs, similar to the effect of reduced growth temperature. Collectively, these data suggest that the proximity complementation assay can identify compounds that improve the plasma membrane traffic of polytopic membrane proteins. 

## 3. Discussion

ABCA4-related retinopathy stands as the most prevalent Mendelian inherited eye disorder, with missense mutations constituting over 60% of the genetic burden [18]. Despite ABCA4 missense variant-associated misfolding being potentially rescuable, little emphasis has been given to the plasma membrane localisation of ABCA4 in vitro or the potential to modify traffic with small molecules and only two publications from the same group have explored this topic so far [40,41]. 

Missense variants in ABCA4 are frequently retained in the ER [13,14]. The ER is the first stage of quality control along the secretory pathway and proteins destined for secretion are co-translationally inserted into the ER lumen, glycosylated and folded [61]. A protein which does not pass quality control assessment will be retained within the ER and subsequently degraded [62]. The ABCA4 missense variants studied in this manuscript show ER retention, presumably by failing ER quality control. The localisation of WT-ABCA4 protein in the ER and/or vesicles could result from the absence in the host cells of the OS or an equivalent structure [63]. The ER resident WT-ABCA4 subpopulation could also be a consequence of the over-expression system, in which the cell machinery gets overwhelmed, and a portion of improperly folded or near-native protein products are not able to traffic towards the vesicle-like structures. 

ABCA4 protein could be detected on the plasma membrane when live cell staining was used on non-permeabilised cells. Nevertheless, only a small proportion of cells had detectable ABCA4 at the plasma membrane and this pattern could not easily be observed in the staining of permeabilised cells, suggesting it is a minor fraction of the total amount of ABCA4. Moreover, the missense variants were not localised on the plasma membrane, supporting their ER retention and impaired traffic towards the cell surface. 

Glycosylation can be used as a surrogate marker for folding and trafficking. For example, CFTR exhibits three distinct forms: the non-glycosylated polypeptide chain (band A), the ER core-glycosylated form (band B) and the fully glycosylated mature form (band C) [64]. In the case of ABCA4, modifying oligosaccharides primarily represent a high-mannose type of N-glycosylation [65]. Consistent with this, treatment with Endo H and PNGase in mouse retina samples resulted in only a slight increase in ABCA4 mobility, suggesting minimal glycan modification in the Golgi (Appendix A) [50]. In HEK293T cells, two ABCA4 species were observed: a likely fully glycosylated band B and an incomplete or non-glycosylated polypeptide band A. This doublet, obtained from RIPA extraction, has been previously described and is known to transform into a sharp, single band considered to represent a native-like structure with the milder CHAPS lysis buffer [16]. RIPA and CHAPS solubilisation have been used as a measure of total expression and as a measure of more native-like proteins, respectively [14]. In our study, further analysis of the doublets revealed a decreased band B/A ratio for the T983A and R2077W ABCA4 variants compared to WT ABCA4, indicating impaired glycosylation of these variants, which could be due to early misfolding events.

Lowering the temperature can positively impact the folding of misfolded proteins, as seen in the functional rescue of the TRPV4 G849A/P851A [66] and the rescue of the traffic of the ΔF508-CFTR [51,52]. We therefore investigated the consequences of reducing temperature on ABCA4 variants’ protein localisation. WT-ABCA4 and missense variant protein levels increased when cells were grown at 30 °C compared to physiological temperature, potentially indicating the increased stability of the protein. Additionally, WT-ABCA4 and missense variants showed an increase in plasma membrane signals. WT-ABCA4 displayed a shift of its ER-resident protein sub-population to the cell membrane under lower temperature conditions. Similarly, the variants escaped the ER to reach the vesicles and/or the membrane. It is likely that lowering the temperature can help the folding of the protein and reduce stress on the cellular folding machinery, allowing the protein to traffic towards the membrane. Whilst changing body temperature is not a practical therapeutic option, the finding highlights that ABCA4 variants are rescuable and suggests that small molecules might be able to correct ABCA4 missense variant folding.

We repurposed the NanoLuc-Binary-based NanoBit protein:protein interaction system (Promega, Madison, WI, USA), which was originally designed to study protein–protein interactions, to produce a proximity complementation assay and quickly and robustly detect plasma membrane protein localisation. This innovative platform falls into the category of protein complementation assays (PCAs) [67]. 

The two nanoluciferase subunits have been optimised in a way that their intrinsic affinity and association constants were outside the ranges of typical interacting proteins [54]. This represents a significant advantage over other technologies, like the fluorescence complementation assays which can produce false positives due to the inherent binding affinity of fragments [67,68]. Even when compared to similar technologies, like the low-affinity β-Gal PCA, the NanoBit assay has other advantages: the minimal fragment used in the low-affinity β-Gal is 46 aa [69], whereas the NanoBit only uses the 11 aa of the SmBiT, possibly causing less impact on the protein of interest’s folding and traffic. This assay is also quick, offering an instantaneous indication of reporter activity in living cells, which is essential for detecting fast interactions or transitory localisation. This is a significant advantage over the bimolecular fluorescence assays, which are characterised by a delay between the interaction of the proteins and the fluorescence readout due to fluorophore maturation [67]. 

Using the NanoBiT assay it was possible to discriminate ABCA4 variants that are retained in the ER and not trafficked to the plasma membrane from control WT-ABCA4 protein that can traffic to the plasma membrane. In particular, T983A, R1108C and R2077W showed significant reductions in luminescence. On the other hand, A1038V had similar levels of luminescence to WT-ABCA4. This variant is usually found in patients as a complex allele with L541P, but taken alone it can traffic to the OS of *Xenopus* PRs [57], suggesting that it is not a misfolding or mistrafficking variant in isolation and that the assay can potentially predict the traffic in PRs. 

We used this platform to screen different small chemical compounds for their ability to enhance ABCA4 traffic to the plasma membrane. Previous studies reported that VX-809 can improve the traffic of WT-ABCA4 and missense variants R1108C, R1129C, A1038V and G1961E [40,41]. However, both A1038V and G1961E have been shown to correctly traffic to the OS of frog and mouse PRs, respectively [57]. We were not able to replicate the effect of VX-809 on the ABCA4 variants in this study. VX-809 and VX-661 have been shown to directly bind to the NBD1 of CFTR in a classic “key in a lock” fashion, stabilising CFTR and making it less susceptible to degradation [70,71]. Their high specificity to CFTR suggests that repurposing these compounds for specific binding to ABCA4 might not be a successful approach. 

The kosmotrope and histone deacetylase (HDAC) inhibitor 4-PBA has found applications in various clinical treatments, including those for cystic fibrosis, thalassemia and urea cycle disorder [72]. FDA-approved for the treatment of urea cycle disorders in children [73], 4-PBA has shown promise in reducing ER retention, preventing misfolding and ameliorating mislocalisation of CFTR mutants both in vitro and in vivo [74,75]. Furthermore, 4-PBA has been shown to rescue the plasma membrane traffic of several missense ABCA1 variants, which share 50.4% of their sequence identity with ABCA4. In this study, we found that 4-PBA can increase ABCA4 variant plasma membrane and vesicular localisation, as well as increasing the steady state level of the proteins. The results mirror the effect of reduced temperature, suggesting that the protein is now able to escape the ER and properly traffic. Further investigations are warranted to discern the mechanism underlying the effect of 4-PBA on ABCA4 and whether the trafficked protein is functional. One plausible hypothesis is that 4-PBA binds to the p24 pocket on COPII, inhibiting the retro-translocation of mutant proteins from the ER to the proteasome [76]. Alternatively, the effect could be mediated by transcriptional changes related to its action as an HDAC inhibitor. 

Another strategy to enhance the folding of mutant proteins involves targeting polypeptide chains early in their synthesis. Mild inhibition of protein translation led to improved mutant CFTR folding and other polypeptides [77,78,79]. The AMPK activators, Metformin and AICAR, have been shown to improve P23H-RHO folding and trafficking in cell culture [45]. In our study, we observed that AICAR boosted ABCA4 plasma membrane trafficking, while the impact of Metformin was not statistically significant. Metformin and AICAR are pleiotropic compounds. Although both are AMPK activators, numerous studies have reported other potential mechanisms. AICAR, in particular, accumulates in millimolar concentrations in cells and exerts many AMPK-independent effects, including beneficial effects on hypoxia, exercise, metabolism and cancer. Regarding their impact on ABCA4, their best known activity is as AMPK activators, slowing down translation rates to enable better folding. However, the different effects in our experiments suggest that AMPK activation is not mediating the effect on ABCA4. The reasons for the different observed effects of AICAR and Metformin on ABCA4 are not entirely clear. 

Taken together these data help shed light on ABCA4 protein biochemistry and cell biology. The NanoBiT assay is quick and robust and can be improved and modified to suit HTS application purposes; however, it has some limitations. Stable cell lines expressing variants of interest would make the screening process easier and faster, enabling a homogenous cell population expressing consistent levels of the protein of interest and subcellular marker. The cell context in which the screening is performed is a very different environment to that experienced by natively expressed ABCA4 in PRs. It will be important to test the lead compounds in a more advanced system recapitulating the cellular mechanisms of ABCA4 folding and traffic in PR-like systems, such as mouse retina or retinal organoids engineered to express ABCA4 variants. Another challenge is lack of a functional assessment. The hypothesis that increasing folding and cell membrane localisation can, subsequently, produce more functional proteins is uncertain, since a “corrected” protein could properly fold and traffic but still exhibit dysfunction. In conclusion, with the appropriate modifications, we believe that this system can be used to make an early assessment of potential compounds affecting ABCA4 behaviour and enable the discovery of compounds for disease causative–missense mutants for ABCA4-related disease.

## 4. Materials and Methods

### 4.1. Immunofluorescence of Cells

Cells were firstly washed two times with phosphate buffered saline (PBS, Gibco, Waltham, MA USA) and then subsequentially fixed with freshly prepared 4% paraformaldehyde (PFA, ThermoFisher, Waltham, MA USA) in PBS for 10 min at room temperature. Samples were blocked and permeabilised with blocking/permeabilising solution containing 10% (*v*/*v*) fetal bovine serum (FBS) and 0.2% Triton X-100 in PBS for 30 min at room temperature. Following this first step, cells were incubated with the primary antibodies (1D4 1:1500, Anti-Calnexin Sigma c4731 1:600, Anti-LgBiT Promega N7100 1:100, Anti-GM130 BD Transduction 610823, Anti-ABCA4 Abcam ab77285 3F4 1:100) for 1 h and 30 min at room temperature in blocking/permeabilisation solution. Cells were incubated for 1 h with the appropriate fluorescently conjugated secondary antibodies (ThermoFisher, all 1:1000) diluted in blocking/permeabilisation solution at room temperature in the dark. Cells were then incubated with 4′,6-diamidino-2-phenylindole (DAPI, Sigma-Alrich, St. Louis, MO, USA) in PBS for nuclear staining. Finally, the slides were mounted with glass coverslips using DAKO fluorescent mounting media (Agilent, Santa Clara, CA, USA) and stored at 4 °C. At least 50 immunoreactive cells were imaged and assessed to determine representative qualitative localisation.

### 4.2. Immunofluorescence of Live Cells

Cells were first washed with PBS and then incubated for 1 h and 30 min at 5% CO_2_ and 37 °C or 30 °C with the primary antibody (Anti-ABCA4 Abbexa abx130549 1:300, Cambridge, UK) diluted in DMEM containing 10% FBS. Samples were then washed 3 times with PBS and incubated in the dark for 1 h at 5% CO_2_ and 37 °C or 30 °C, with the appropriate fluorescently conjugated secondary, always diluted in DMEM with 10% FBS, added. The cells were then fixed with freshly prepared 4% PFA for 10 min and, from this step onward, the immunofluorescence was carried on following the same protocol as described in the previous paragraph. 

### 4.3. Cell Protein Extraction Using RIPA Buffer

Cold PBS was used to wash the cells and RIPA buffer supplemented with 2% (*v*/*v*) protease inhibitor cocktail (Sigma) and 1% phosphatase inhibitor cocktail (Sigma) was used to lyse the cells for at least 30 min at 4 °C. The cell lysates were then collected and sonicated for 10 s. Following the sonication, the samples were centrifuged using a cold centrifuge at 4 °C for 30 min at 30,000× *g* to remove cellular debris. After the centrifugation, the supernatant was collected and the precipitate was discarded. Protein levels were measured with the Pierce^TM^ BCA protein assay kit (ThermoFisher) according to the manufacturer’s instructions.

### 4.4. Western Blot

Following protein extraction and quantification, Laemmli sample buffer 5× containing 250 mM Tris-HCl pH 6.8, 50% (*v*/*v*) Glycerol, 10% (*w*/*v*) SDS, 525 mM DTT and Bromophenol Blue was added to each lysate and 6 or 10 μg proteins were resolved via SDS-PAGE on 8% or 10% acrylamide gels. Proteins were electrophoretically transferred onto a 0.45 μm Polyvinylidene difluoride (PVDF) membrane via wet transfer using transfer buffer 1× Tris-glycine with 20% (*w*/*v*) ethanol. Membranes were then blocked in 5% (*w*/*v*) non-fat dried milk in PBS-Tween (PBST, 0.1% (*w*/*v*) Tween-20 in PBS) for one hour at room temperature. Membranes were then incubated overnight with primary antibodies (1D4 1:1000 (cordially obtained from Prof. Rober S. Molday) (Figure 3 and Figure 4), Vinculin 4650S Cell Signalling 1:1000 (Danvers, MA, USA), GAPDH Proteintech 60004 1:10,000 (Rosemont, IL, USA), Tubulin Sigma T4026 1:2000, ABCA4 Abcam ab72955 1:1000 (Cambridge, UK) (Figure 7 and Figure 8)) diluted in 5% (*w*/*v*) milk-PBST at 4 °C in agitation. The following day the membranes were incubated with the secondary anti-mouse or anti-rabbit antibody coupled with horseradish peroxidase 1:30,000 (Invitrogen, Carlsbad, CA, USA) in 5% (*w*/*v*) milk-PBST. Following the incubation, the membranes were washed 3 times for 10 min each and then developed with the Clarity or Clarity Max ECL Western Blotting Substrates (Bio-Rad, Hercules, CA, USA). The chemiluminescence detection was performed using the Chemidoc MP system (Bio-Rad) and the signal intensity was measured on ImageJ.

### 4.5. Endo H and PNGase-F Digestion

The glycosylation status of the proteins was studied using the PNGase-F and the Endo H (NEB) glycosidases according to the manufacturer’s instructions. The samples were then studied by western blotting. 

### 4.6. Protein Translation Inhibition 

Cycloheximide (CHX) was used to inhibit protein synthesis. Cells were treated with 50 μM of CHX (Sigma-Aldrich, St. Louis, MO, USA) in culture medium 48 h after transfection, for 2 or 4 h, at 37 °C and 5% CO_2_. Cells were then lysed, processed for protein extraction and studied by western blotting. 

### 4.7. Luminescence Protocol and Detection 

Luminescence signals were detected following the NanoBit Protein:Protein Interaction System manufacturer’s protocol. Luminescence was detected using a luminometer (Berthold detection system, Baden-Württemberg, Germany) with detection timing set between 20 to 5/s per well. 

### 4.8. Plasmid Production 

NEB 5-alpha competent *E. coli* cells (#C2988, NEB) or NEB 5-alpha competent *E. coli* cells (High Efficiency, #C2987) were used to perform bacterial transformation according to the manufacturer’s instructions. The ZymoPURE^TM^ (Cambridge-Bioscience, Cambridge, UK) plasmid midiprep kit was used from 50 mL bacterial cultures to isolate plasmid DNA according to the manufacturer’s guidance. ABCA4 variant plasmids were provided by Professor Bob Molday. RP2 was added to pBiT1.1-C[TK/LgBiT] by site-directed-mutagenesis. Forward and reverse primers were produced using NEBaseChanger^TM^ (New England Biolabs, Ipswich, MA, USA) *RHO* and *RHO-P23H* were cloned into the pBiT2.1-C[TK/SmBiT] to produce RHO-SmBiT and P23H-SmBiT. WT-ABCA4 was assembled into the pBiT2.1-C[TK/SmBiT] plasmid using NEBuilder HiFi DNA assembly technology according to the manufacturer’s instructions to produce WT-ABCA4-SmBiT. ABCA4 missense variants were introduced in WT-ABCA4-SmBiT by site-directed mutagenesis using a Q5 Site-Directed Mutagenesis Kit quick protocol (NEB, E0554) according to the manufacturer’s instructions (Appendix A). Forward and reverse primers were produced using an NEBaseChanger^TM^. The sequences of the WT-ABCA4-SmBiT and RP2 LgBiT are shown in (Appendix A). 

### 4.9. Cell Culture and Transient Transfection of DNA Plasmids

Cells were cultured in Dulbecco’s Modified Eagle Medium (DMEM GlutaMAX^TM^, Gibco) supplemented with 10% (*v*/*v*) FBS, penicillin (100 U/mL) and streptomycin (100 μg/mL). In general, CHO cells were used for immunofluorescence purposes since they possess well-defined cytoplasm that enables clearer imaging for presentation of figures. On the other hand, HEK293T were used for protein lysates due to their high transfection efficiency and the high amount of heterologous proteins they can produce by enhancing signal to noise. Cells were transfected with *Trans*IT^®^-LT1 Transfection Reagent 24 h after reaching 60–70% confluency. *Trans*IT-LT1 Reagent and plasmids DNA were diluted in Opti-MEM reduced-serum medium in a sterile tube and left to incubate for 30 min at room temperature. The mixture was then gently added to the cells. 

## Figures and Tables

**Figure 1 ijms-25-04521-f001:**
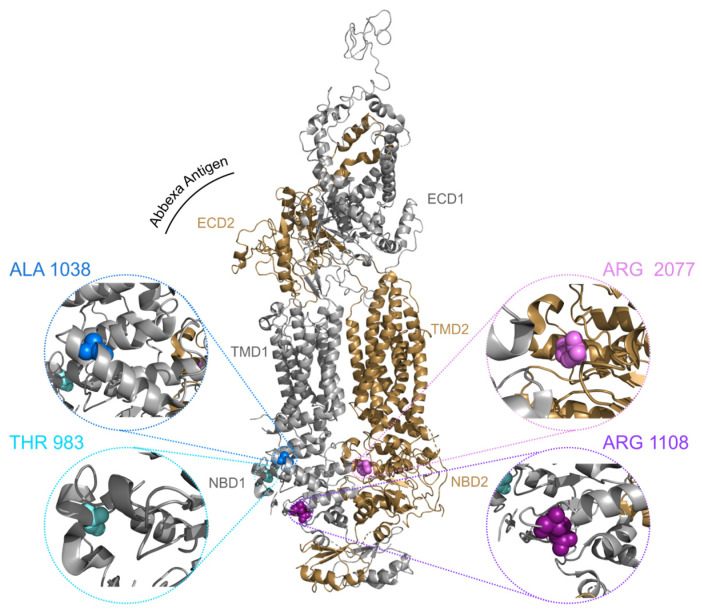
**Structure of ABCA4 in the unbound state.** The structure is represented as a cartoon with the N- and C-terminal halves coloured in silver and gold, respectively. The selected residues are represented as spheres and colour coded. The structure has been modelled in Pymol (The PyMOL Molecular Graphics System, Version 2.5.5 Schrödinger LLC, New York, NY, USA) using the cryo-EM map of substrate-free ABCA4 [42]. The α-ABCA4 Abbexa antibody antigen is highlighted with a black curve.

**Figure 2 ijms-25-04521-f002:**
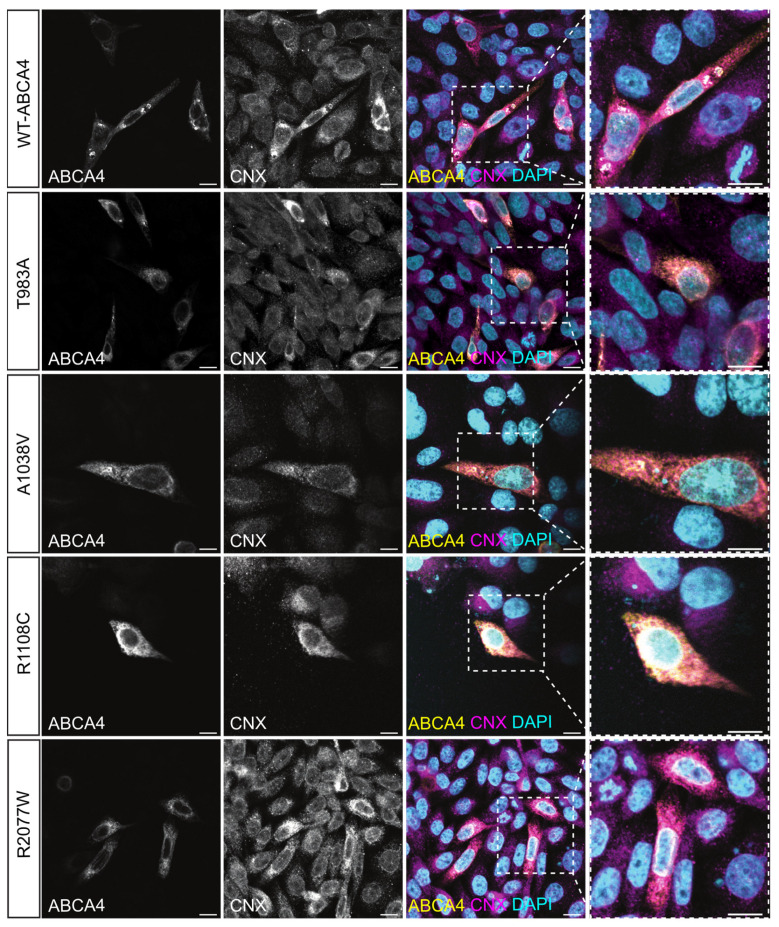
**Localisation of WT ABCA4 and disease-associated variants.** CHO cells were transfected with plasmids expressing WT-ABCA4 protein and the indicated variants. At 48 h post-transfection, cells were fixed in 4% PFA and permeabilised with 0.2% Triton X-100 and labelled with 1D4 antibody (yellow) and CNX antibody (magenta). Nuclei were stained with DAPI (cyan). WT-ABCA4 mostly localised in large CNX-positive intracellular vesicle-like structures, whereas the indicated mutants localised in a reticular pattern characteristic of the ER (insets). Scale bars = 10 μm.

**Figure 3 ijms-25-04521-f003:**
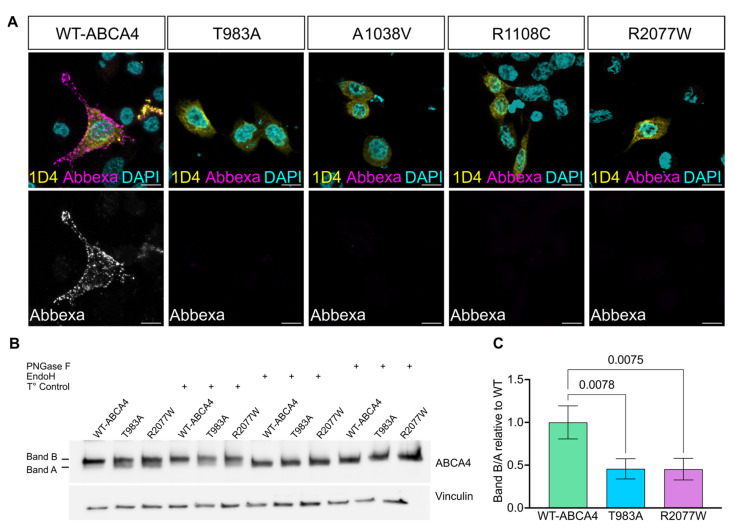
**Missense ABCA4 variants present partially impaired trafficking and maturation in cultured cells.** (**A**) CHO cells were transfected with plasmids expressing WT-ABCA4 and selected mutants. At 48 h post-transfection, cells were double-labelled with the 1D4 antibody to stain the intracellular ABCA4 (yellow) and with the ABCA4 Abbexa antibody (magenta, upper panels; white, lower panels) to stain the membrane-exposed extracellular epitope. Nuclei were stained with DAPI (cyan). WT-ABCA4 was localised both in the ER and at the plasma membrane, whereas missense variants were mainly trapped in the ER. Scale bars = 10 μm. (**B**) HEK293T cells were transfected with plasmids expressing WT-ABCA4, T983A and R2077W. At 48 h post-transfection, cell lysates were collected, and 6 µg of protein lysate was treated with glycosidases Endo H and PNGase-F, or buffer and temperature-only protocol control (T°). The untreated sample migrated as two species, band A and band B. The treated samples only showed band A. (**C**) The B/A band ratio was quantified using ImageJ (Version 2.14.0/1.54f). Fold changes relative to WT are shown. T983A and R2077W band ratios were reduced with respect to WT protein. Error bars are mean of fold change ± SD. n = 3 independent experiments. One-way ANOVAs and post-hoc analysis comparisons were performed only against WT ABCA4 sample.

**Figure 4 ijms-25-04521-f004:**
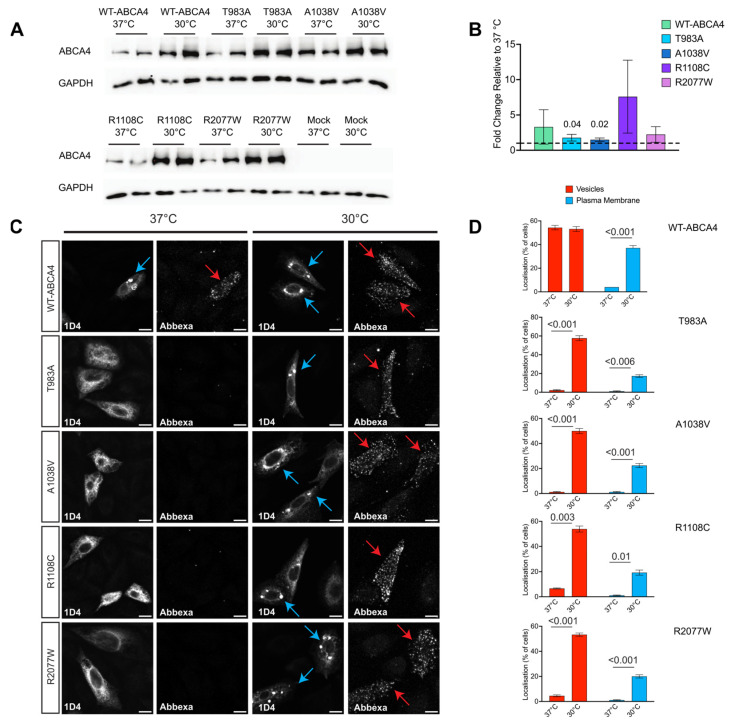
**The effect of temperature on ABCA4 protein level and localisation.** (**A**) HEK293T cells were transfected with plasmids expressing WT-ABCA4 and the indicated variants and subsequently incubated at 37 °C or 30 °C. WT ABCA4 and variant protein levels detected by western blot (10 µg of protein lysate). Mock represents a non-transfected control. (**B**) Quantification of band intensity using ImageJ. Data were normalised to GAPDH reference protein intensity and 37 °C level (dashed line, 1.00). Error bars are mean ± SD. n = 3 independent experiments. Significant differences are displayed. (**C**) CHO cells transfected with WT-ABCA4 and variants incubated at 37 °C or 30 °C and stained for cell surface accessible ABCA4 (Abbexa) and total ABCA4 (1D4). Confocal images showing the localisation of ABCA4 in vesicle structures (blue arrows) and plasma membrane (red arrows). Cells treated at 30 °C show an increase in the plasma membrane and vesicle-positive cells. (**D**) ABCA4 vesicle-positive cells and plasma membrane-positive cells were scored among all transfected cells and expressed as a percentage of total transfected cells. n = 10 fields of view (×40 objective). Mean ± SD. Scale bars = 10 μm. Two-tailed Student’s *t*-test for 37 °C vs. 30 °C.

**Figure 5 ijms-25-04521-f005:**
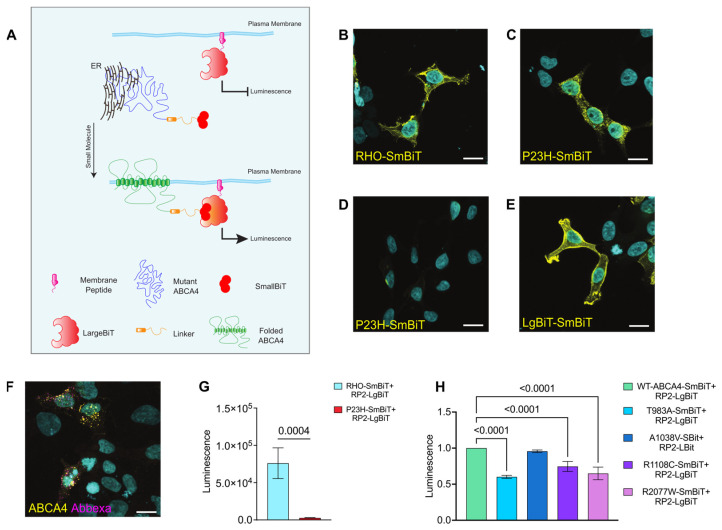
**Split NanoBiT proximity complementation assay**. (**A**) Schematic showing the design of the system. Misfolded membrane protein (ABCA4; blue) is tagged at the cytoplasmic C-terminus with SmBiT part of NanoLuc (dark red) and the LgBiT subunit (light red) is targeted to the plasma membrane by the N-terminus of RP2 (magenta). When misfolded protein is retained in the ER the SmBiT (dark red) and LgBiT (light red) cannot complement (**upper panel**). Small molecule rescue of ABCA4 plasma membrane traffic enables the SmBiT–LgBiT system to produce luminescence through proximity induced complementation of the NanoLuc (**lower panel**). (**B**) WT-RHO fused with SmBiT localises to the plasma membrane. The staining was performed with antibody 4D2 without a permeabilization step, as the 4D2 epitope is located on the extracellular N-terminus the protein. (**C**) P23H fused to SmBiT staining is observed in a perinuclear and reticular pattern, consistent with the ER, following a detergent permeabilization step to reveal 4D2 immunoreactivity. (**D**) P23H-SmBiT is not detected with 4D2 using a no-permeabilization protocol. (**E**) The RP2-LgBiT localised to the plasma membrane, when stained with anti-LgBiT antibody in permeabilized cells. (**F**) Intracellular WT-ABCA4-SmBiT was detected with the ABCA4 3F4 ABCAM antibody (yellow) and WT-ABCA4-SmBiT localising to the plasma membrane was detected using an Abbexa antibody (magenta) 48 h post-transfection analysis. Scale bars = 10 μm. (**G**) 24 h post-transfection with RHO-SmBiT + RP2-LgBiT and P23H-SmBiT + RP2-LgBiT plasmids luminescence signal was measured. Raw luminescence values are shown. Error bars are ± SD. n = 3. two-tailed Student’s *t*-test for RHO vs. P23H. (**H**) Luminescence signal was analysed 48 h post-transfection with ABCA4 variants + RP2-LgBiT plasmids in live HEK293T cells. Mean of fold change relative to WT ± SD. n = 4 independent experiments. One-way ANOVA and post-hoc analysis comparisons were performed only against WT-ABCA4-SmBit + RP2-LgBit sample. Significant differences are displayed.

**Figure 6 ijms-25-04521-f006:**
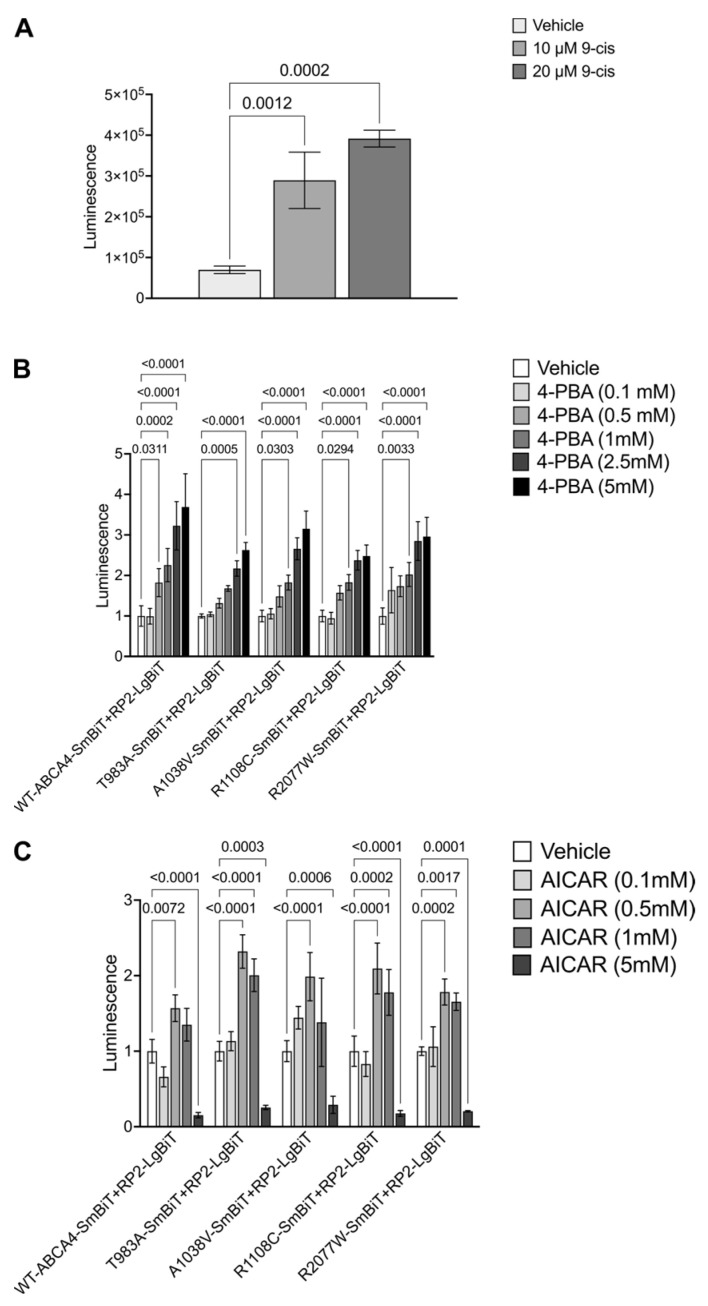
**Cell-based complementation assay reveals drug potency for cell surface traffic.** (**A**) HEK293T cells were treated with 9-cis-retinal for 24 h, and luminescence was analysed 48 h post-transfection. P23H-SmBiT + RP2-LgBiT treated with 10 µM and 20 µM of pharmacological chaperone showed a significant increase in luminescence. Raw values are shown ± SD, n = 3. One-way ANOVAs and post-hoc analysis comparisons were conducted only against vehicle. (**B**,**C**) At 48 h post transfection, cells were treated with different concentrations of 4-PBA (**B**) and AICAR (**C**) for 24 h. Luminescence signal was measured. Fold change relative to the vehicle ± SD. n = 3. Two-way ANOVA against WT-ABCA4-SmBit + RP2-LgBit sample. Significant differences after post-hoc correction analysis are displayed.

**Figure 7 ijms-25-04521-f007:**
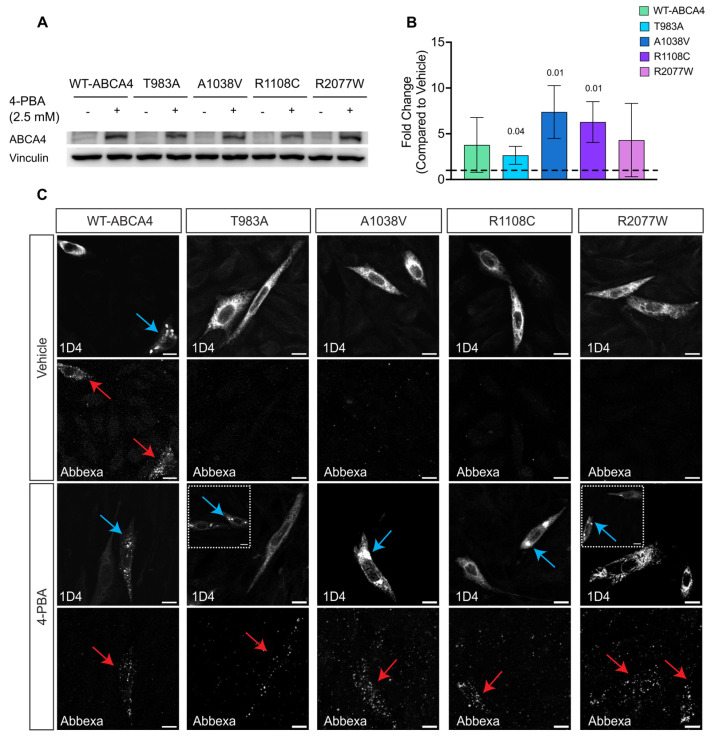
**The effect of 4-PBA on ABCA4 protein expression and traffic.** (**A**,**B**) Western blot analysis and quantification of ABCA4 variants protein level (10 µg protein lysate), mean fold change ± SD compared to vehicle after normalisation to vinculin, n = 3. Black dashed line represents baseline vehicle value, 1.00. Two-tailed Student’s *t*-test for treated vs. vehicle. Significant differences are displayed. (**C**) CHO cells transfected with ABCA4 variants were treated with 5 mM 4-PBA for 24 h. The sub-cellular localisation was analysed by confocal microscopy using two different antibodies recognising ABCA4 intra-(1D4) and extra-(Abbexa) cellular epitopes. Blue arrows indicate vesicles, red arrows indicate ABCA4 at the plasma membrane, and, for T983A and R2077W, the insets show cells with vesicular staining. Scale bars = 10 μm.

**Figure 8 ijms-25-04521-f008:**
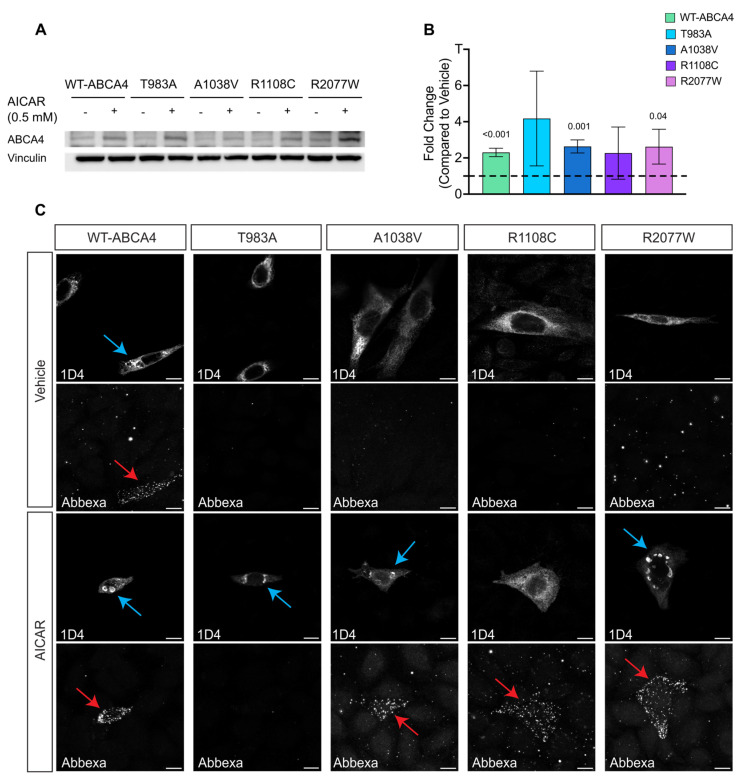
**The effect of AICAR on ABCA4 protein.** (**A**,**B**) Western blot analysis and respective quantification of 10 µg of protein lysate from HEK293T cells expressing ABCA4 variants showing that AICAR treatment increases the steady state of the ABCA4 protein. n = 3. Error bars are mean of fold change ± SD. Black dashed line represents baseline vehicle value, 1.00. Two-tailed Student’s *t*-test for treated vs. vehicle. Significant differences are displayed. (**C**) CHO cells transfected with ABCA4 variants were treated with 0.5 mM AICAR for 24 h. The sub-cellular localisation was analysed by confocal microscopy using two different antibodies recognising ABCA4 intra-(1D4) and extra-(Abbexa) cellular epitopes. Blue arrows indicate vesicles, red arrows indicate ABCA4 at the plasma membrane. Scale bars = 10 μm.

## Data Availability

The data that support the findings of this study are available from the corresponding authors, M.E.C. and D.P.

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
