# Peer review of "A Proximity Complementation Assay to Identify Small Molecules That Enhance the Traffic of ABCA4 Misfolding Variants"

_ijms, 2024, doi:10.3390/ijms25084521_

Round 1
Reviewer 1 Report
Comments and Suggestions for Authors
In this manuscript, Piccolo and colleagues describe a novel method to screen ABCA4 traffic to the plasma membrane as a potential target of new drugs. ABCA4 missense variants were transfected in cells, ABCA4 membrane traffic was assessed and ultimately, a proximity assay was conducted to study ABCA4 membrane localization.
Major comments
1. The lack of an endogenous control where ABCA4 is expressed is a concern. This reviewer understands the complexity, but I overall recommend the authors to circumvent that issue.
2. It is not clear whether the authors performed positive (i.e. a housekeeping gene and calculating the fold increase expression of it) and negative controls (i.e. mock target) for the transfection protocols. mRNA quantification matching the protein expression in WT and missense groups by western blot would be also appreciated.
3. In lines #104 and 105 and again in lines #387 to 389, the authors mention that ABCA4 is retained in the ER. Have the authors calculated the co-localization rate of ABCA4 and calnexin? “Retention” should be associated with any form of quantification in comparison to the protein signal outside of the ER.
4. Ideally, missense variants and WT sequences insertions should be sequenced.
Specific comments
1. Results 2.1: I suggest the authors provide the percentage of cells effectively transfected for each variant and N of cells analyzed.
2. Figure 2B: Vinculin control loading looks uneven in the main text and also in “Uncropped Figures- Figure 2” file.
3. I suggest moving Supplementary figure 5 to the main Figure 4 to illustrate the model.
4. Line #369-371: lacking reference or, if it is related to not published data.
5. Include WT and missense variants sequences in methods or supplementary data.
Reviewer 2 Report
Comments and Suggestions for Authors
The manuscript entitled A proximity complementation assay to identify small molecules that enhance the traffic of ABCA4 misfolding variants is presented for the peer review. ABCA4-related-retinopathy is the most common inherited Mendelian eye disorder worldwide, caused by biallelic variants in the ATP-binding cassette transporter ABCA4. ieved through a therapeutic approach based on folding correction using chemical chaperones, pharmacological chaperones, or altering molecular chaperone activity. These approaches can rescue defects in the folding of mutant proteins by providing stabilization 60 during the folding of the protein, thereby enabling the correct conformational maturation. Currently no approved therapies target ABCA4. In this study, authors demonstrated possibility that ABCA4 misfolding variants are temperature-sensitive and reduced temperature growth (30°C) improves their traffic to the plasma membrane, suggesting the folding of these variants could be rescuable. Consequently, an in vitro platform was developed for the rapid and robust detection of ABCA4 traffic to the plasma membrane in transiently transfected cells. The manuscript is well-written and structured. Literature list contains up-to-date sources. Authors did a lot of elegant experiments to prove clinical significance of their work.
I have several suggestions for authors.
1. Please illustrate protein sequences to show mutant sites.
2. Please provide explanation of cell -specific effect of mutant protein localization. DO you think it;s enough to use only 2 cell lines?
3. Have you implied rescue experiment to revert cell phenotype p.5 lines 165-174?
Minor corrections: please add size bars to all Figures.
Comments on the Quality of English LanguageEverything is fine.
Reviewer 3 Report
Comments and Suggestions for Authors
The topic discussed in this paper is a logic continuation of the research field where deviated temperature can influence final protein shape and function. Unlike the topic discussed in recent Nature communications paper https://doi.org/10.1038/s41467-024-46332-6 where high temperatures showed decreased evolution of protein phenotype this paper focuses on the known fact that lower temperatures may lead to better protein formation process decreasing abnormalities of the final proteins even if they carry mutated domains.
In cases where misfolded ABCA4 proteins cannot be efficiently refolded or degraded, they may aggregate together with other misfolded proteins, forming inclusion bodies within the ER. These inclusion bodies can disrupt ER function and contribute to cellular stress and dysfunction.
In normal circumstances, properly folded ABCA4 proteins are transported from the ER to the Golgi apparatus and subsequently to their final destinations within the cell membrane. However, misfolded proteins may have impaired trafficking, leading to retention within the ER or mislocalization to other cellular compartments.
The fate of misfolded ABCA4 protein in the cell is influenced by the interplay between (1) cellular quality control mechanisms, (2) chaperone-mediated folding pathways, and the (3) severity of the protein misfolding defect. Understanding these processes is crucial for elucidating the molecular mechanisms underlying ABCA4-related retinal diseases and developing potential therapeutic interventions.
Results of this research may be applied as a possible treatment in the future after exploring pharmacological agents that help to stabilize the proteins and go through cellular quality control mechanisms. This approach showed its efficacy in cystic fibrosis treatment due to CFTR gene mutations.
The work done by the research team is immense, robust results are presented.
Comments and questions:
1. I would like to see some more explanation why CHO cell line was chosen for the part of experiments along with HEK293T cells.
2. Do authors have explanation why R1108C mutation (shown in purple) in ABCA4 shows highest difference in folding at 30C comparing to 37C? Fig3B.
Minor comments:
Line 43: byproduct instead of biproduct
I recommend to publish this research after minor revision.
Round 2
Reviewer 1 Report
Comments and Suggestions for Authors
I thank the authors for accepting my suggestions. All potential issues/limitations were properly addressed in the author's response. With that, I will recommend the manuscript for publication.